# Transcriptome Sequence Reveals Candidate Genes Involving in the Post-Harvest Hardening of Trifoliate Yam *Dioscorea dumetorum*

**DOI:** 10.3390/plants10040787

**Published:** 2021-04-16

**Authors:** Christian Siadjeu, Eike Mayland-Quellhorst, Shruti Pande, Sascha Laubinger, Dirk C. Albach

**Affiliations:** Institute for Biology and Environmental Sciences, Carl-von-Ossietzky University Oldenburg, Carl-von-Ossietzky Str. 9-11, 26111 Oldenburg, Germany; eike.mayland.quellhorst@uol.de (E.M.-Q.); shruti.pande@uol.de (S.P.); sascha.laubinger@uol.de (S.L.); dirk.albach@uol.de (D.C.A.)

**Keywords:** *D. dumetorum*, yam, tuber, orphan crop, post-harvest hardening, transcriptome, RNA-Seq, gene expression

## Abstract

Storage ability of trifoliate yam (*Dioscorea dumetorum*) is restricted by a severe post-harvest hardening (PHH) phenomenon, which starts within the first 24 h after harvest and renders tubers inedible. Previous work has only focused on the biochemical changes affecting PHH in *D. dumetorum*. To the best of our knowledge, the candidate genes responsible for the hardening of *D. dumetorum* have not been identified. Here, transcriptome analyses of *D. dumetorum* tubers were performed in yam tubers of four developmental stages: 4 months after emergence (4MAE), immediately after harvest (AH), 3 days after harvest (3DAH) and 14 days after harvest (14DAH) of four accessions (Bangou 1, Bayangam 2, Fonkouankem 1, and Ibo sweet 3) using RNA-Seq. In total, between AH and 3DAH, 165, 199, 128 and 61 differentially expressed genes (DEGs) were detected in Bayangam 2, Fonkouankem 1, Bangou 1 and Ibo sweet 3, respectively. Functional analysis of DEGs revealed that genes encoding for *CELLULOSE SYNTHASE A* (*CESA*), *XYLAN O-ACETYLTRANSFERASE* (*XOAT*), *CHLOROPHYLL A/B BINDING PROTEIN*
*1*, *2*, *3*, *4* (*LHCB1*, *LHCB2*, *LHCB3*, and *LCH4*) and an *MYB* transcription factor were predominantly and significantly up-regulated 3DAH, implying that these genes were potentially involved in the PHH as confirmed by qRT-PCR. A hypothetical mechanism of this phenomenon and its regulation has been proposed. These findings provide the first comprehensive insights into gene expression in yam tubers after harvest and valuable information for molecular breeding against the PHH.

## 1. Introduction

Yams constitute an important food crop for over 300 million people in the humid and subhumid tropics. Among the eight yam species commonly grown and consumed in West and Central Africa, trifoliate yam (*Dioscorea dumetorum*) is the most nutritious [1]. Tubers of *D. dumetorum* are rich in protein (9.6%), well balanced in essential amino acids (chemical score of 0.94) and its starch is easily digestible [2,3]. *Dioscorea dumetorum* is not only used for human alimentation but also for pharmaceutical purposes. A bio-active compound, dioscoretine, has been identified in *D. dumetorum* [4], which has been accepted pharmaceutically and which can be used advantageously as a hypoglycemic agent in situations of acute stress. The tubers are, therefore, commonly used in treating diabetes in Nigeria [5].

Despite these qualities, the storage ability of this yam species is restricted by severe post-harvest hardening (PHH) of the tubers, which begins within 24 h after harvest and renders them unsuitable for human consumption [1]. The PHH of *D. dumetorum* is separated into a reversible component associated with the decrease of phytate and an irreversible component associated with the increase of total phenols [6]. The mechanism of PHH is supposed to start with enzymatic hydrolyzation of phytate and subsequent migration of the released divalent cations to the cell wall where they cross-react with demethoxylated pectins in the middle lamella. This starts the lignification process in which the aromatic compounds accumulate on the surface of the cellular wall reacting as precursors for the lignification [7].

Whereas physiological changes associated with the hardening of yam tubers are now reasonably well-understood, knowledge is still limited in terms of ways to overcome the hardening. Naturally, occurring genotypes lacking PHH have been identified [8], which offers a chance to understand the genetic basis of hardening. Therefore, the next step is to understand the genetic background of this genotype and its relationship to other genotypes, which has been conducted using GBS (Illumina-based genotyping-by-sequencing [9]. Further insights have been gained by sequencing and analyzing the genome of the non-hardening genotype Ibo sweet 3 [10].

Here, we analyze the transcriptome of the non-hardening accession Ibo sweet 3 and three hardening accessions to identify genes involved in the PHH phenomenon. The study of the transcriptome examines the presence of mRNAs in a given cell population and usually includes some information on the concentration of each RNA molecule, as a factor of the number of reads sequenced, in addition to the molecular identities. Unlike the genome, which is roughly fixed for a given cell line when neglecting mutations, the transcriptome varies from organ to organ, during development and based on external environmental conditions. In particular, transcriptome analysis by RNA-seq enables the identification of genes that have differential expression in response to environmental changes or developmental stages and mapping genomic diversity in non-model organisms [11]. Differential gene expression analysis under different conditions has, therefore, allowed an increased insight into the responses of plants to external and internal factors and into the regulation of different biological processes. High-throughput sequencing technologies allow an almost exhaustive survey of the transcriptome, even in species with no available genome sequence [12]. Indeed, transcriptome analysis based on high-throughput sequencing technology has been applied to investigate gene expression of hardening in carrot [13]. In yam, it helped elucidate flavonoid biosynthesis regulation of *D. alata* tubers [14].

A lack of availability of next-generation ‘–omics’ resources and information had hindered the application of molecular breeding in yam [15], which has recently been overcome by the publication of two genome sequences in the genus [10,11,12,13,14,15,16]. Here, we report the first transcriptomic study of *D. dumetorum* and the first to evaluate the influence of genes on the PHH phenomenon in a monocot tuber using transcriptomics. We aim to close this gap by identifying candidate genes involved in the PHH phenomenon of *D. dumetorum* to facilitate breeding non-hardening accessions of *D. dumetorum*.

## 2. Results

### 2.1. Descriptive Statistics of RNA-Seq Data

Transcriptome sequences of four *D. dumetorum* accessions (Bangou 1, Bayangam 2, Fonkouankem 1 and Ibo sweet 3) were analyzed during four tuber developmental stages: 4 months after emergence (4MAE), 9 months after emergence (harvest time, AH), 3 days after harvest (3DAH) and 14 days after harvest (14DAH) to determine putative genes involved in the PHH phenomenon. After trimming, 943,323,048 paired-end raw reads (150-bp in length) were generated for 48 samples (Appendix A). Among these, 242.7, 224.6, 233.9 and 242.1 million reads belonged to Bangou 1, Bayangam 2, Fonkouankem 1 and Ibo sweet 3. On average, 90% of all clean reads were aligned to the *D. dumeotrum* reference genome v1.0. Furthermore, an average of 56% of those reads was uniquely mapped to the reference genome sequence. A principal component analysis (PCA) plot showed the normalized read counts of all samples (Figure 1). The first two principal components (PCs) explained 69% of the variability among samples. Samples four months after emergence were separated from those AH, 3DAH and 14DAH. No clear separation was observed between AH, 3DAH, and 14DAH. However, taking into account accession specificity, AH was separated from 3DAH and 14DAH. This finding indicated a difference in transcriptome expressions of accessions before and after harvest. One biological replicate of each accession at a specific time point did not cluster with others likely due to individual variability between plants.

### 2.2. Differential Expression Analysis

Two well-established statistical analysis methods (edgeR and DESeq2) to assess differentially expressed genes (DEGs) based on reading counts were employed. We used two strategies to determine DEGs in *D. dumetorum* after harvest: STAR_DESeq2, and STAR_edgeR. The design model for DE analysis was ~ Accession + Condition + Accession:Condition. We carried out multiple comparisons at the accession, condition and interaction accession * conditions levels. The STAR_DESeq2 approach yielded the highest number of DEGs (Figure 2) and the results were selected for downstream analysis (Appendix A). Pairwise comparisons (4MAE vs. AH, 3DAH vs. AH, 14DAH vs. AH, 14DAH vs. 3DAH) of gene expressions among the four accessions were performed (Figure 2). However, since the PHH in *D. dumetorum* tubers occurs after harvest, we focused on gene expressions after harvest. A decrease of up-regulated DEGs and an increase of down-regulated DEGs were noticed among the three accessions that do harden from harvest to 14DAH (Figure 2). The accession that does not harden depicted a different pattern. Comparing 3DAH vs. AH, significantly DEGs were detected in Bayangam 2 (165 DEGs), Fonkouankem 1 (199 DEGs), Bangou 1 (128 DEGs) and Ibo sweet 3 (61 DEGs). Amongst these, 120, 112, 83 and 16 were up-regulated in Bayangam 2, Fonkouankem 1, Bangou 1 and Ibo sweet 3, respectively. For 14DAH vs. AH, significantly DEGs were obtained in Bayangam 2 (162 DEGs), Bangou 1 (201 DEGs), Fonkouankem 1 (161 DEGs) and Ibo sweet 3 (46 DEGs). Among these, 126, 83, 47, and 13 were up-regulated DEGs in Bayangam 2, Bangou 1, Fonkouankem 1 and Ibo sweet 3, respectively. In total, the highest number of significantly up-regulated DEGs were detected in Bayangam 2 and the lowest in Ibo sweet 3. A mixture analysis of the three accessions that do harden irrespective of accession was performed (Appendix A). Pairwise comparisons of gene expression among the three stages or conditions (AH, 3DAH and 14DAH) in the combined analysis of the three hardening accessions detected 59, 40 and 13 up-regulated DEGs between 3DAH vs. AH, 14DAH vs. AH and 14DAH vs. 3DAH, respectively (Appendix A). Whereas, 14 (3DAH vs. AH), 36 (14DAH vs. AH), and 56 (14DAH vs. 3DAH) were down-regulated.

In order to understand the difference between Ibo sweet 3 (the non-hardening accession) and the other accessions, a multiple pairwise comparison (Bayangam 2 vs. Ibo sweet 3, Bangou 1 vs. Ibo sweet 3, Fonkouankem 1 vs. Ibo sweet 3) after harvest (3DAH vs. AH, 14DAH vs. AH) was carried out (Figure 3, Appendix A). After harvesting 3DAH (3DAH vs. AH), significantly, DEGs were acquired comparing Bayangam 2 vs. Ibo sweet 3 (111 DEGs), Fonkouankem 1 vs. Ibo sweet 3 (111 DEGs) and Bangou 1 vs. Ibo sweet 3 (80 DEGs). Amongst these, 101, 80 and 62 were up-regulated DEGs in Bayangam 2 vs. Ibo sweet 3, Fonkouankem 1 vs. Ibo sweet 3 and Bangou 1 vs. Ibo sweet 3, respectively. For 14DAH vs. AH, 88, 85 and 91 significantly DEGs were detected comparing Bayangam 2 vs. Ibo sweet 3 (88 DEGs), Fonkouankem 1 vs. Ibo sweet 3 (85 DEGs) and Bangou 1 vs. Ibo sweet 3 (91 DEGs). Among these, 80, 30 and 22 were up-regulated in Bayangam 2 vs. Ibo sweet 3, Fonkouankem 1 vs. Ibo sweet 3 and Bangou 1 vs. Ibo sweet 3, respectively.

### 2.3. GO Enrichment and Functional Classification of DEGs with KEGG and Mapman

For better comprehension of the PHH phenomenon, gene ontology (GO) term annotation and enrichment were performed on up-regulated DEGs resulting from pairwise comparisons (3DAH vs. AH, 14DAH vs. AH) of all three accessions that do harden (Figure 4A). Compared with 3DAH and AH, out of the 59 up-regulated DEGs, 38 were significantly annotated with 43 GO terms, most of which were involved in biological processes related to cellular processes, response to stimuli, and metabolic processes. Likewise, for 14 DAH vs. AH, 23 up-regulated genes (out of 40) were significantly enriched regarding biological processes in relation to cellular processes, response to stimuli, and metabolic processes (Figure 4B). GO term analysis of each hardening accession separately revealed that cellular processes, metabolic processes, response to stimuli and response to stress were in the top 10 of the most common enriched GO terms 3DAH and 14DAH (Figure 4C,D).

Pathway-based analysis with Kyoto encyclopedia of genes and genomes (KEGG) revealed that metabolic pathway (Ko01100) was the most enriched with seven and six up-regulated transcripts followed by biosynthesis of secondary metabolites pathway (Ko01110) with three and one up-regulated transcripts 3DAH and 14DAH, respectively (Figure 5A,B). Based on MapMan, photosynthesis (Bin 1, 23 genes) and RNA biosynthesis pathways (Bin 15, eight genes) were the most enriched 3DAH. Likewise, 14DAH, photosynthesis (six genes) and RNA biosynthesis (six genes) were the most enriched pathways (Figure 5A,B).

Analysis of each hardening accession separately showed a similar pattern for KEGG and MapMan pathway-based annotations (Figure 5C,D). The metabolic pathway was the most enriched followed by biosynthesis of secondary metabolites pathway 3DAH and 14 DAH. Fourteen (in Bayangam 2), twelve (in Fonkouankem 1), and nine (in Bangou 1) up-regulated transcripts related to metabolic pathways were recovered 3DAH. Conversely, fifteen, five, and three up-regulated transcripts were identified in Bayangam 2, Fonkouankem 1, and Bangou 1 respectively 14DAH. The MapMan pathway enrichment revealed that photosynthesis, RNA biosynthesis and cell wall organization pathways were in the top 5 of the most common enriched pathway across the hardening accessions 3DAH. However, protein homeostasis, phytohormone action, and protein modification were among the top 5 of the most common enriched pathways 14DAH. These results highlight that the PHH likely occurs predominantly in *D. dumetorum* 3DAH.

### 2.4. Cluster Expression Analysis

Based on the differential expression analysis and functional annotation results, the PHH likely occurs 3DAH. Thus, differentially expressed up-regulated genes 3DAH in the hardening accessions and in the combined analysis of the three hardening accessions together were selected for cluster expression analysis. Their expressions during the four tuber development stages were plotted (Figure 6). This analysis aims to identify groups of genes with similar expression patterns in all hardening accessions in relation to PHH. Two groups or clusters were identified amongst up-regulated DEGs 3DAH (Figure 6). The first pattern depicted a high peak 4MAE and then decreased AH and slightly increased 3DAH and 14DAH with an expression under zero except for the accession Fonkouankem 1. This first pattern corresponds to cluster 1 in Bangou 1 and Fonkouankem 1 and cluster 2 in Bayangam 2 and in the mixture of the three hardening accessions (Figure 6A–D). Conversely, for the second pattern, the expression was down 4MAE and AH, and sharply increased 3DAH and then decreased 14DAH. This second pattern corresponds to cluster 2 in Bangou 1 and Fonkouankem 1 and cluster 1 in Bayangam 2 and the mixture of the three hardening accessions. This second pattern, showing the highest peak 3DAH could be the group of genes that are co-regulated and involved in the PHH. Therefore, functional annotation of genes of clusters showing the second pattern, was further investigated.

The top 3 accumulated pathways in cluster 2 were photosynthesis (20 contigs) followed by solute transport (two contigs) and cell wall organization (one contig) in Bangou 1 (cluster 2) (Appendix A). For Bayangam 2 (cluster 1), the top 3 pathways were protein modification (eight contigs) followed by RNA biosynthesis (seven contigs) and phytohormone action (seven contigs). However, it is worth outlining that cell wall organization (four contigs) and secondary metabolism (three contigs) pathways were as well accumulated. On the contrary, in Fonkouankem 1 (cluster 2), cell wall organization (19 contigs) was the most enriched pathway followed by RNA biosynthesis (eight contigs) and photosynthesis, secondary metabolism, protein homeostasis, cytoskeleton organization and solute transport pathways with four contigs each of them. The mixture of the three hardening accessions (cluster 1) showed that photosynthesis was the most accumulated pathway (21 contigs) followed by protein homeostasis, lipid metabolism pathways with three contigs each of them and cell wall organization pathway with two contigs. In sum, genes encoding for photosynthesis, cell wall organization, protein modification and RNA biosynthesis, and secondary metabolism pathways are co-up-regulated after harvest and likely involved in the PHH on *D. dumetorum* tubers.

### 2.5. Comprehensive Analysis of Expression of Genes Potentially Involved in PHH

We opted for the investigation of genes differentially expressed 3DAH in the accession Fonkouankem 1 (cluster 2) due to its high amount of up-regulated genes associated with cell organization and the combined analysis of the three hardening accessions together (cluster 1). In the mixture of the three hardening accessions, (cluster 1), a total of 20 transcripts homologous to the genes encoding for photosynthesis were observed as up-regulated differentially expressed three 3DAH, when all hardening accessions were analyzed together (Table 1), including *CHLOROPHYLL A/B BINDING PROTEINS LHCB1* (eight transcripts), *LHCA4* (two transcripts) *LHCB2* (two transcripts), *PHOTOSYSTEM II PROTEIN PSBX* (two transcripts). Those genes respond to light stimulus and may be the triggers of this phenomenon. Three transcripts associated with cell wall organization were found encoding for *FASCICLIN-TYPE ARABINOGALACTAN PROTEIN*, *CORNCOB CELLULOSE* (*COB*) and *GLUCAN ENDO-1,3-BETA-GLUCOSIDASE***.** They are likely involved in the reinforcement of the cell wall (hardening)**.** One transcript homologous to the gene related to *MYOLOBLASTTOSIS* (*MYB*) transcription factors (TFs) was included in this group. However, it is important to note that genes involved in lipid metabolism, namely LIPASE (three transcripts) were found in this group.

In Fonkouankem 1 (cluster 2) (Table 2), 18 up-regulated genes encoding for cell wall organization including *XYLAN O-ACETYLTRANFERASE* (*XOAT*) (five transcripts), *CELLULOSE SYNTHASE* (*CESA*) (three transcripts), *COB* (two transcripts) were found in cluster 2. The *MYB* transcription factor was the most abundant (four transcripts) followed by TFs *DREB* and *NAC* with 2 transcripts each of them. Photosynthesis genes *LHCB1*, and *LHCA4* were found with two transcripts each of them. However, genes encoding for phenolic metabolism were enriched with two genes *CINNAMATE 4-HYDROXYLASE* (two transcripts) and *PHENYLALANINE AMMONIA LYASE* (two transcripts). Likewise, *LIPASE* (three transcripts) was recorded in this group.

In all hardening accessions and the combined analysis of the three hardening accessions together, annotation with several *MYB* database identified putative *MYB* genes (*MYB54*, *MYB52*, *MYB73*, *MYB70*, *MYB44*, *MYB77*, *MYB46*, *MYB83*, *MYB9, MYB107*, *MYB93*, *MYB53*, and *MYB92*) associated with cell wall modifications (Appendix A).

### 2.6. Comprehensive Difference between Harden and Non-Harden Accessions

Pairwise comparisons of accessions that do harden to the accession that does not harden in different stages after harvest showed that up-regulated genes were enriched mostly in cellular process, cellular anatomical entity and intracellular terms 3DAH and 14DAH (Figure 7, Appendix A). Besides, KEGG enrichment revealed that metabolic pathways were the most enriched with ten, eight and five up-regulated genes 3DAH for Bayangam 2 vs. Ibo, Fonkouankem vs. Ibo and Bangou 1 vs. Ibo, respectively (Figure 7A–C). This pathway was followed by biosynthesis of secondary metabolites pathway with six, five, and five up-regulated genes for Bayangam 2 vs. Ibo, Fonkouankem 1 vs. Ibo sweet 3 and Bangou 1 vs. Ibo sweet 3 respectively. Those pathways were the most enriched as well 14DAH (Appendix A). MapMan annotation showed that the cell wall organization pathway was predominantly enriched when comparing Bangou 1 to Ibo sweet 3 and Fonkouankem 1 vs. Ibo sweet 3 3DAH. Whereas, protein modification pathway was particularly enriched for Bayangam 2 vs. Ibo sweet 3. However, cell organization, protein modification and RNA biosynthesis pathways were in the top 7 of the most enriched pathways 3DAH. On the contrary, the protein modification pathway was the most enriched irrespective of the comparison 14DAH (Appendix A). The Venn diagram of the annotation revealed five common up-regulated genes potentially involved in the hardening process among the accessions that do harden comparing to the non-hardening accession Ibo sweet 3. Those genes encoding for *CHALCONE SYNTHASE*, *DITERPENE SYNTHASE*, an *MYB* transcription factor, *XOAT*, *LIGNIN LACCASE* (Figure 7D).

Four up-regulated and co-regulated DEGs (*LHCB1*, *CESA*, *LACCASE*, and *MYB46*) involved in the PHH were selected for validation in two development stages (AH and 3DAH) in the accession Fonkouankem 1 using quantitative reverse transcription-polymerase chain reaction (qRT-PCR). The expression levels in each biological replicate of all these genes substantially increased from AH to 3DAH (Figure 8). Thus, the four DEGs identified from the transcriptome analysis were significantly up-regulated 3DAH (AH vs 3DAH). These results were consistent with the transcriptome data analysis.

## 3. Discussion

The PHH of *D. dumetorum* tubers has been extensively studied in terms of the biochemical and physical aspects [1,2,3,4,5,6,7,8,9,10,11,12,13,14,15,16,17]. Based on our study we reported genes that were differentially expressed and up-regulated after harvest. This demonstrates that the PHH on *D. dumetorum* tuber is likely to be genetically regulated. Our results demonstrate that the number of up-regulated genes was abundant 3DAH and then decreased 14DAH. This suggests that PHH predominantly occurs in the first days after harvest. This is consistent with previous studies [1,8,18] showing a substantial increase of the hardness in the first three DAH.

Functional analysis via KEGG enrichment revealed that most genes differentially expressed were involved in pathways of secondary metabolites. These genes are involved in photosynthesis, RNA biosynthesis (transcription factors), and cell wall organization. In order to understand how this phenomenon occurs, GO enrichment revealed that many genes were involved in cellular processes, response to stimuli, and metabolic processes, as well as response to stress. These results firmly suggest that PHH in *D. dumetorum* is a cellular and metabolic process in response to stimuli leading to stress.

It has been reported that PHH in *D. dumetorum* is associated with an increase in sugar and structural polysaccharides (cellulose, hemicellulose, and lignin) [1]. Later, Medoua et al. [18] associated it with a decrease of phytate and total phenols. However, these authors failed to address the causes of this phenomenon. Cellular processes are triggered by a stimulus, an investigation of genes related to response to stimuli revealed that photosynthetic genes *LHCB1,2,3* and *LCH4* were up-regulated 3DAH. Those genes are light-harvesting chlorophyll a/b binding antenna responsible for photon capture. This suggests that *D. dumetorum* tubers are starting photosynthesis after harvest. In the field, *D. dumetorum* tubers turn green under the yam skin (on the surface) when they are exposed to sunlight (Appendix A). Unlike potatoes, greening occurs only in the field but not in storage. This highlights the importance of water in this process. After harvest, tubers are exposed to the external environment with no possibility of water absorption. This likely leads to drought stress as supported by GO term analysis. In fact, a rapid decrease of water in tubers after harvest was reported [1,2,3,4,5,6,7,8,9,10,11,12,13,14,15,16,17,18], probably mainly due to the photosynthetic activity of *D. dumetorum* tubers. Thus, the PHH of *D. dumetorum* tubers appears to be a mechanism to limit water loss.

Mechanisms to limit water loss in plants have been extensively associated with the reinforcement of the cell wall [19]. Medoua et al. [18] reported a decrease in water absorption by tubers after harvest suggesting that the cell wall permeability decreases during storage. Genes related to cell wall organization *XOAT*, *CESA*, *COB* were predominantly up-regulated after harvest. This confirms biochemical changes associated with the PHH of *D. dumetorum* tubers [1,2,3,4,5,6,7,8,9,10,11,12,13,14,15,16,17,18]. An increase in various cell wall polysaccharide such as cellulose, hemicellulose and lignin during storage has been reported [1,2,3,4,5,6,7,8,9,10,11,12,13,14,15,16,17,18]. Cellulose synthase encodes for cellulose biosynthesis [20] and *COB* regulates the orientation of cellulose microfibrils, whereas, *XOAT* encodes for hemicellulose (xylan) biosynthesis [21]. These cell wall polysaccharides play an important role as a protective barrier in response to various environmental perturbations. Accumulation and deposition of these polysaccharides inside primary cell walls reinforce the strength and rigidity of the cell wall and are probably a key component of the plant response to environmental factors [19]. It suggests that cellulose and lignin are key cell wall polymers responsible for cell wall rigidification during PHH in *D. dumetorum*.

Many biological processes are controlled by the regulation of gene expression at the level of transcription. Transcription factors are key players in controlling cellular processes. Among those TFs, the *MYB* family is large and involved in controlling diverse processes such as responses to abiotic and biotic stresses [22]. Our results demonstrated that TF from the *MYB* family was predominantly expressed and up-regulated after harvest. This result suggests that transcription factors from the *MYB* family may be potentially involved in the mechanism of PHH. Guo et al. [23] demonstrated the role of an *MYB* TF family in response to water stress from the stem of a birch tree through lignin deposition, secondary cell wall thickness and the expression of genes in secondary cell wall formation.

A pairwise comparison of the hardening accessions and the non-hardening accession confirmed that the PHH phenomenon is a cellular and metabolic process leading to cell wall modification. However, it is interesting to note that protein modifications seem to occur predominantly after hardening from 3 to 14DAH. This could explain the poor sensory qualities of hardened tubers such as coarseness in the mouth [24]. Five common genes were found up-regulated in the hardening accessions and down-regulated in Ibo sweet 3 3DAH. Those genes are *CHALCONE SYNTHASE, DITERPENE SYNTHASE*, transcription factor *MYB*, *XYLAN O-ACETYLTRANSFERASE* and *LIGNIN LACCASE*. CHALCONE SYNTHASE is a key enzyme of the flavonoids/isoflavonoid biosynthesis pathway and is induced in plants under stress conditions [25]. LACCASE catalyzes the oxidation of phenolic substrates using oxygen as an electron acceptor. Laccase has been recognized in the lignification process through the oxidation of lignin precursors. Indeed, Arcuri et al. [26] demonstrated involvement of *LACCASE* genes in lignification as a response to adaptation to abiotic stresses in *Eucalyptus*.

Based on our results, the PHH seems to be governed by differentially expressed genes in a metabolic network, which is attributed to exposure to the external environment or sunlight. Therefore, a putative model of the hardening mechanism and the regulatory network associated was proposed (Figure 9). After harvest, yam tubers are exposed to the external environment particularly to sunlight. This environmental factor acts as the first signal to stimulate the expression of photosynthetic genes involved in photon capture namely *LHCB1*, *LHCB2*, *LHCB3* and *LHCA4*. The absorption of photons implies loss of electrons which is replaced by electrons from the splitting of water through photolysis [27]. This activity implies the necessity of a continued electron supply through the breakdown of water molecules. However, tubers are detached from roots with no possibility of water absorption. Therefore, a signal is given to reinforce the cell wall to avoid water loss from the tubers via the up-regulation of *CESA*, *XOAT* and *COB* genes. This reinforcement of the cell wall implies firstly, an accumulation of cell wall polysaccharide such as cellulose hemicellulose during the first days of storage. Secondly, probably from the third day after harvest, the lignification process controlling laccase genes begins. This overall process is might be controlled by an *MYB* TF. This mechanism was further validated by qRT-PCR of the main genes involved in the PHH, showing the reliability of our data.

## 4. Materials and Methods

### 4.1. Plant Materials

Four accessions were collected from various localities in the main growing regions of yam (West and South-West) in Cameroon and one from Nigeria based on the analysis of [9]. These accessions were characterized by many roots (Bangou 1, Bayangam 2, Fonkouankem 1) and few roots (Ibo sweet 3) on the tuber surface with yellow flesh color. Among these accessions, tubers of Ibo sweet 3 do not harden after harvest. Ten tubers of each accession were planted in pots in the greenhouse of the botanic garden of the University of Oldenburg under controlled conditions at 25 °C. They are available upon request.

### 4.2. Sample Preparation

Three tubers of each accession were randomly collected 4 months after emergence (4MAE), 9 months after emergence (Harvest time AH), 3 days after harvest (3DAH) and 14 DAH. After harvest, tubers were stored under shelter in the greenhouse at 25 °C and light intensity of 1.76 µmol s^−1^ m^−2^. Collected tubers were washed and their skin peeled off. Then, the samples were immediately frozen in liquid nitrogen and stored at −80 °C prior to RNA isolation.

### 4.3. RNA-Seq Extraction

The stored tubers (−80 °C) were lyophilized prior to further handling. Total RNA was extracted from 48 samples [(Four accessions (Bangou 1, Bayangam 2, Fonkouankem 1, Ibo sweet 3) × four stages (4MAE, AH, 3DAH, and 14DAH) × three biological replicates)] using innuPREP Plant RNA Kit (Analytik Jena AG, Jena, Germany). The RNA quality was analyzed using a spectrophotometer (Nano-Drop Technologies, Wilmington, DE, USA). RNA Integrity Number (RIN) values were determined using a Bioanalyzer 2100 (Agilent Technologies, Santa Clara, CA, USA) to ensure all samples had an RNA integrity number (RIN) above 6.

### 4.4. Library Construction and Illumina Sequencing

We constructed cDNA libraries comprising 48 RNA samples using the Universal Plus mRNA-Seq offered by NuQuant (Tecan Genomics, Inc., Redwood City, CA, USA). Paired-end (2 × 150 bp) sequencing of the cDNA libraries was performed on an Illumina NovaSeq (Genewiz Germany GmbH, Leipzig).

### 4.5. Data Processing and Functional Analysis

Low-quality reads were filtered using TrimGalore v 0.6.5 (https://github.com/FelixKrueger/TrimGalore/releases, accessed on 15 April 2021) with the following parameters—length 36—q 5—stringency 1 × 10^−1^–e 0.1. The filtered reads were aligned to the reference genome of *D. dumetorum* [10] with STAR v 2.7.3a [28] with default parameters. The aligned reads in BAM files were sorted and indexed using SAMtools v 1.9 [29]. The number of reads that can be assigned uniquely to genomic features was counted using the function SummarizeOverlaps of the R package GenomicAlignments v1.20.1 [30] with mode = “Union”, singleEnd = FALSE, ignore.strand = TRUE, fragments = TRUE as parameters.

Two programs DESeq2 [31] and edgeR [32] were deployed to analyze differentially expressed genes (DEGs) between conditions and the interaction conditions × accessions. Gene with *p*-adjusted value < 0.05 and log2 fold change > 2 were considered as significantly expressed genes. False discovery rate FDR threshold was <0.05. We performed a basic time-course experiment to assess genes that change their expression after harvest using Deseq2 [31]. Metabolic pathway assignments of DEGs were based on the KEGG Orthology database using the KAAS system [33]. The final pathway analyses were mostly based on the tools Mercator4 and Mapman4 [34]. In addition, differentially-expressed *MYB* genes were functional annotated based on several datasets *Arabidopsis thaliana MYB*s [35], *Beta vulgaris MYB*s [36], *Musa acuminata MYB*s [37], *Croton tiglium MYB*s [38], *Dioscorea rotundata MYB*s and *Dioscorea dumetorum MYB*s via KIPEs (https://github.com/bpucker/KIPEs, accessed on 15 April 2021). GO term assignment and enrichment were performed using Blast2GO [39] via OmicsBox with cutoff 55, Go weight 5, e-value 1.e-6, HSP-hit coverage cutoff 80 and hit filter 500.

### 4.6. Cluster Differential Expressed Genes Analysis

Co-expression analysis was carried out to identify up-regulated genes with similar expression patterns using The k-means cluster method. Genes that are clustered together across different time points and conditions may be involved in the same regulatory processes [40]. The number of clusters was determined through the sum of squared error and the average silhouette width as described in this script (https://2-bitbio.com/2017/10/clustering-rnaseq-data-using-k-means.html, accessed on 15 April 2021).

### 4.7. qRT-PCR Validation of Targeted Gene

Transcriptomic data were validated with four DEGs involved in PHH using qRT-PCR analysis. RNA extractions were performed using RNeasy Plant Mini Kit (Qiagen Sciences, Germantown, MD, USA) from yam tubers of the Accession Fonkouankem 1 AH and 3DAH. To remove excess starch, total RNA was cleaned up and concentrated using RNA Clean and Concentrator Kit (Zymo Research Europe GMBH, Freiburg im Breisgau, Germany). Total RNA was then treated with DNase I (Thermo Scientific, Vilnius, Lithuania) and cDNA synthesis was carried out using oligo(dT) primers and RevertAid First Strand cDNA Synthesis Kit (Thermo Scientific, Germany). qRT-PCR was carried out with 1 μL of cDNA in a total volume of 10 μL using Maxima SYBR Green (Thermo Scientific, Schwerte, Germany) on CFX384 Real-Time PCR system (Biorad Laboratories. Inc., Munich, Germany). Primers were designed using Primer-BLAST (Primer3web ver 4.1.0) [41] (Appendix A) and primer efficiencies were determined by serial dilution of cDNA. Three biological replicates including controls without template were used in all qPCR reactions. Relative expressions were calculated using the 2(−∆∆CT) method using ACTIN as control.

## 5. Conclusions

This is the first study that investigates differentially expressed genes after harvest and during yam storage through RNA-Seq. The evidence from this study suggests that PHH in *D. dumetorum* is a cellular and metabolic process involving a combined action of several genes as a response to environmental stress from sun and water. Genes encoding for cell wall polysaccharide constituents were found significantly up-regulated suggesting that they are directly responsible for the hardening of *D. dumetorum* tubers. It is worth noticing that many genes encoding for light-harvesting chlorophyll a/b binding proteins were significantly up-regulated after harvest as well. This supports the idea that sunlight is the trigger element of the PHH manifested by the strengthening of cell walls in order to avoid water loss, which is useful for a putative photosynthetic activity. These findings add substantially to our understanding of hardening in *D. dumetorum* and provide the framework for molecular breeding against PHH in *D. dumetorum*.

## Figures and Tables

**Figure 1 plants-10-00787-f001:**
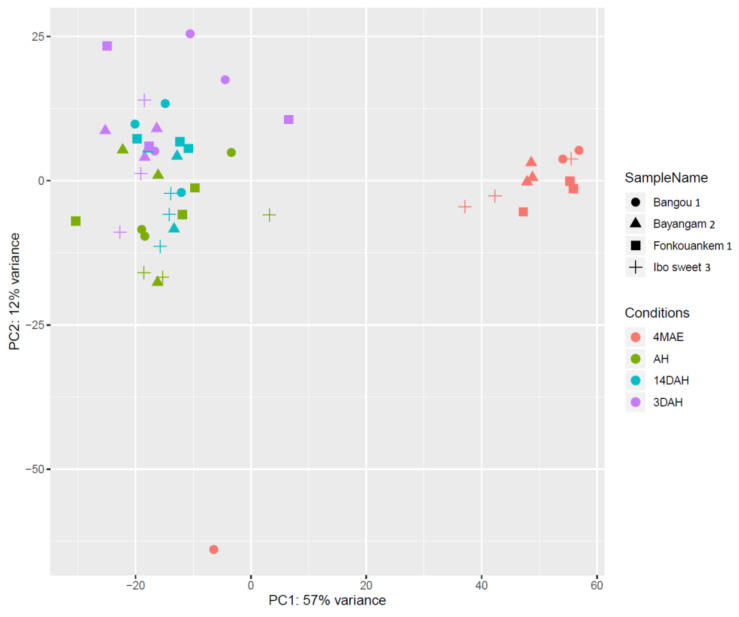
PCA plot of normalized counts using the variant stabilizing transformation (VST). Four *D. dumetorum* accessions (symbols) at four different sampling points (colors) are shown.

**Figure 2 plants-10-00787-f002:**
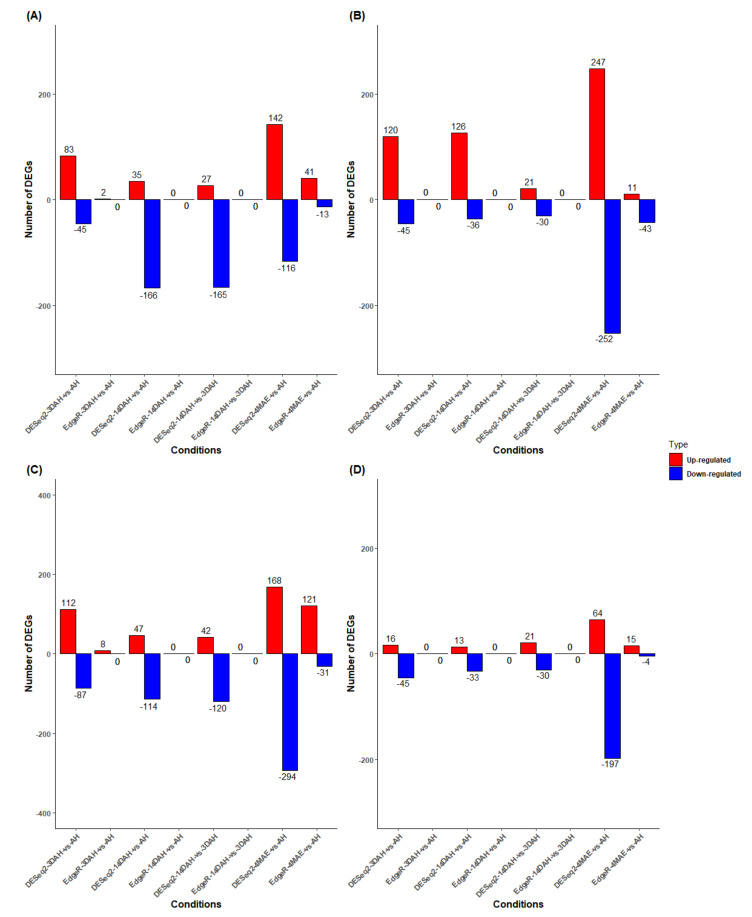
The number of differentially expressed genes (DEGs) based on the comparison of DESeq2 and EdgeR 4MAE and after harvest (AH, 3DAH, and 14DAH). (**A**) Bangou 1, (**B**) Bayangam 2, (**C**) Fonkouankem 1, (**D**) Ibo sweet 3 (non-hardening accession). Blue represents down-regulated transcripts and red represents up-regulated transcripts.

**Figure 3 plants-10-00787-f003:**
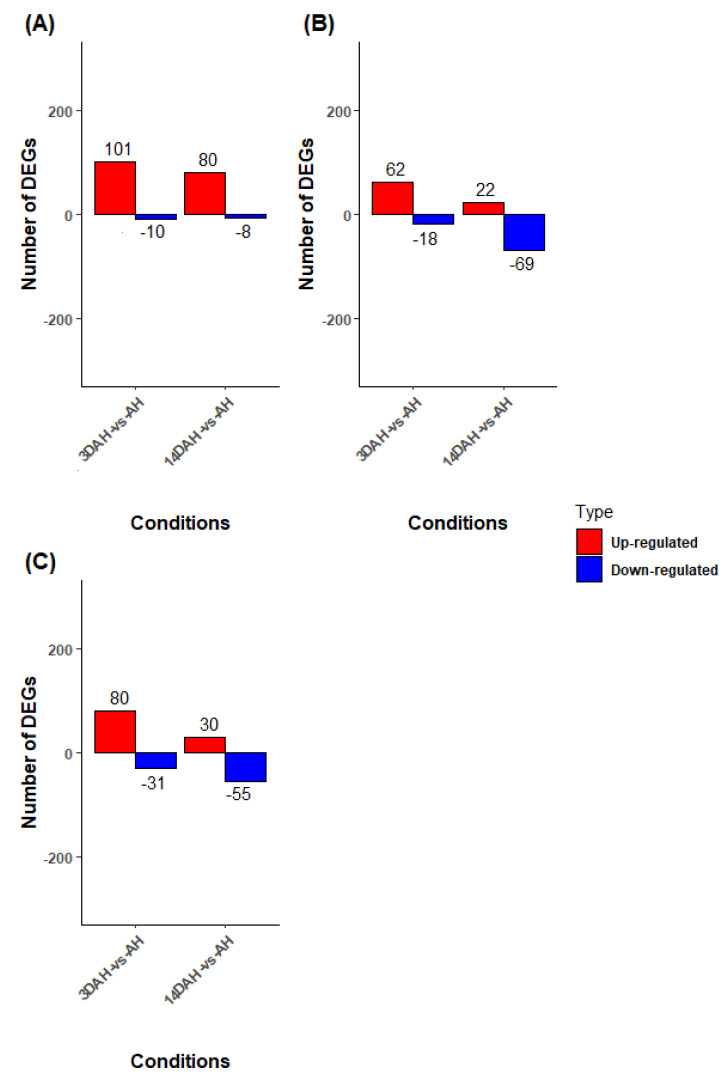
The number of DEGs based on the comparison between Ibo sweet 3 and other accessions after harvest (AH, 3DAH, and 14DAH). (**A**) Ibo sweet 3 vs. Bayangam 2, (**B**) Ibo sweet 3 vs. Bangou 1, (**C**) Ibo sweet 3 vs. Fonkouankem 1. Blue represents down-regulated transcripts and red represents up-regulated transcripts.

**Figure 4 plants-10-00787-f004:**
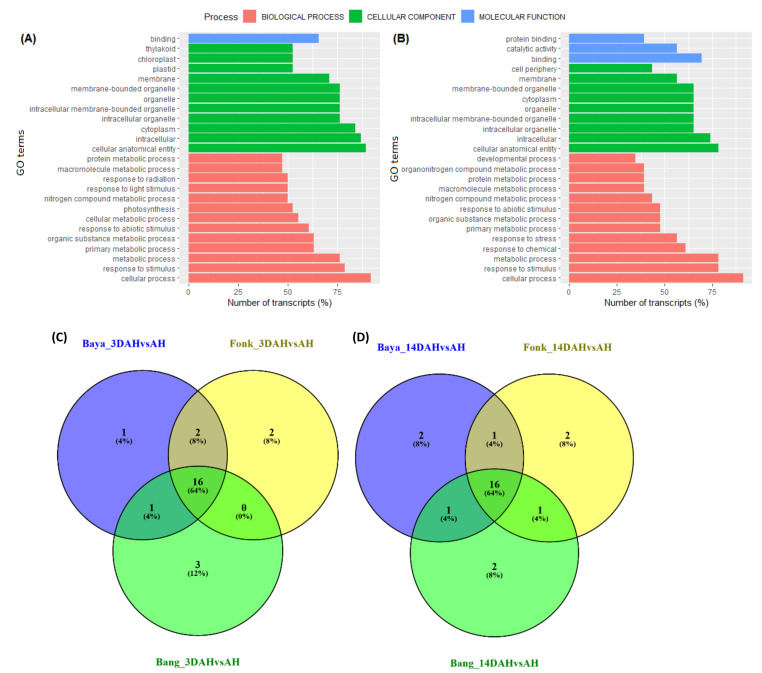
Functional annotation of the top up-regulated enriched gene ontology (GO) terms of *D. dumetorum* tubers after harvest (AH, 3DAH, 14DAH). (**A**,**B**) the top 25 up-regulated enriched GO terms of the combined analysis of three hardening accessions 3DAH and 14DAH, respectively. (**C**,**D**) Venn diagrams of the enrichment of the top 20 GO terms of each hardened accession 3DAH and 14DAH, respectively. The blue bar represents a molecular process, the green bar represents the cellular component, and red bar represents the biological process.

**Figure 5 plants-10-00787-f005:**
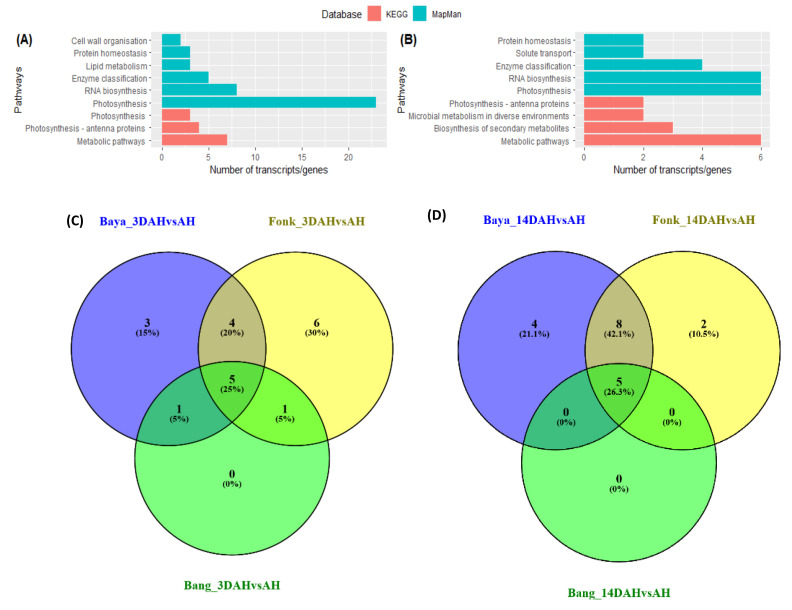
Functional classification of DEG after harvest. (**A**,**B**) the most enriched pathways of the combined analysis of three hardening accessions 3DAH and 14DAH, respectively. (**C**,**D**) Venn diagrams the most enriched pathways of each hardening accession 3DAH and 14DAH respectively. Green bars represent pathway annotation with MapMan data-base, and red bars represent pathway annotation with the Kyoto encyclopedia of genes and genomes (KEGG) database.

**Figure 6 plants-10-00787-f006:**
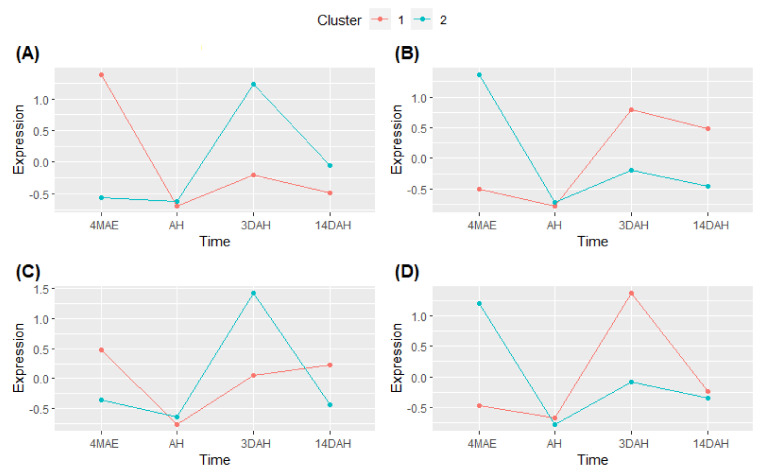
Cluster analysis of DEGs 3DAH among the different sampling times 4MAE and after harvest. (**A**) Bangou 1, (**B**) Bayangam 2, (**C**) Fonkouankem 1, (**D**) combined analysis of the three hardening accessions.

**Figure 7 plants-10-00787-f007:**
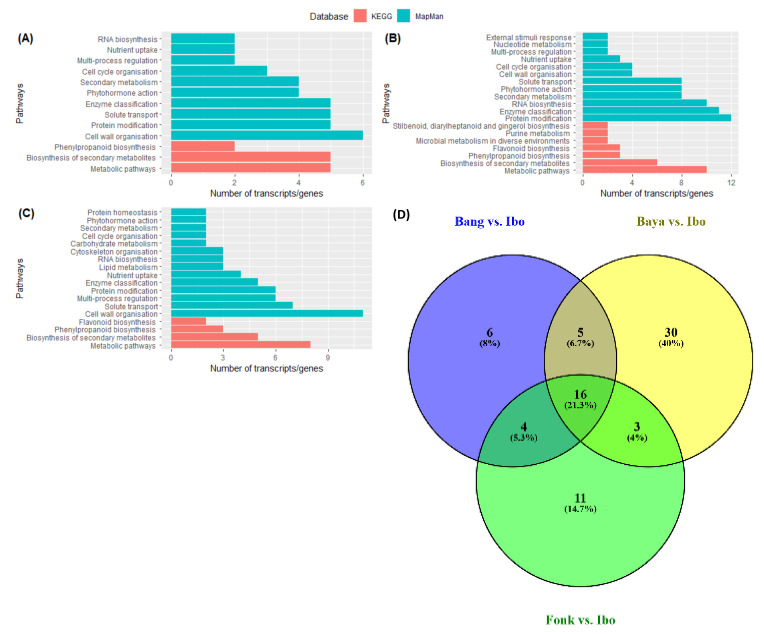
Functional classification of up-regulated DEG 3DAH based on the comparison of hardened accessions against the non-hardening accessions. (**A**–**C**) the most enriched pathways 3DAH on Bangou 1 vs. Ibo sweet 3, Bayangam 2 vs. Ibo sweet 3 and Fonkouankem 1 vs. Ibo sweet 3, respectively. (**D**) Venn diagram of the most enriched pathways on Bangou 1 vs. Ibo sweet 3, Bayangam 2 vs. Ibo sweet 3 and Fonkouankem 1 vs. Ibo sweet 3. Green represents pathway annotation with MapMan database, and red represents pathway annotation with the KEGG database.

**Figure 8 plants-10-00787-f008:**
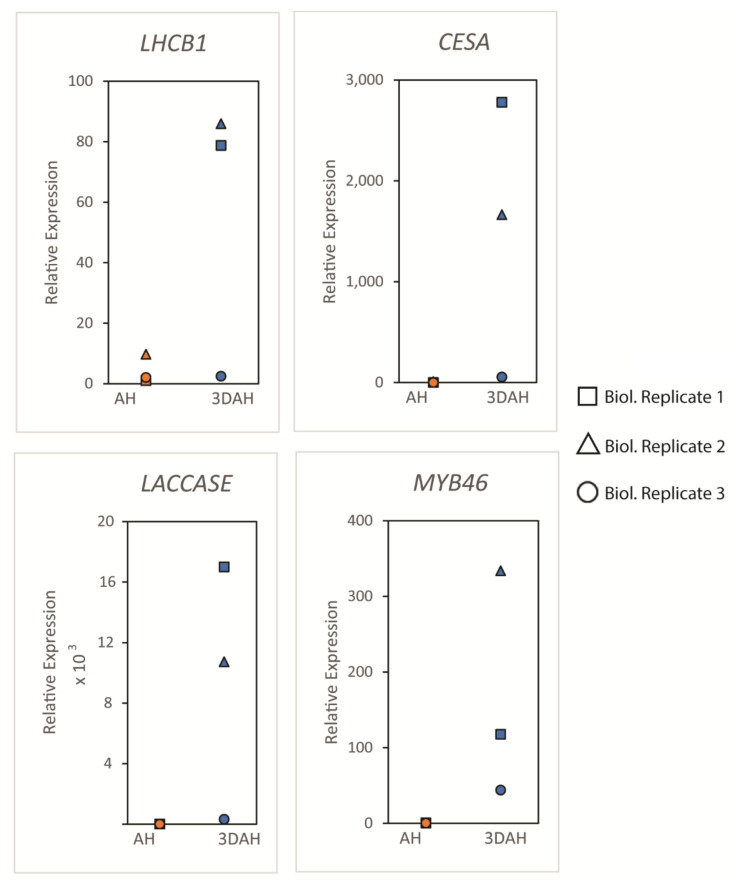
qRT-PCR analysis of *LHCB1*, *CESA*, *LACCASE* and *MYB46* on 3 biological replicates of yam tubers in Fonkouankem AH and 3DAH. Shapes represent each biological replicate.

**Figure 9 plants-10-00787-f009:**
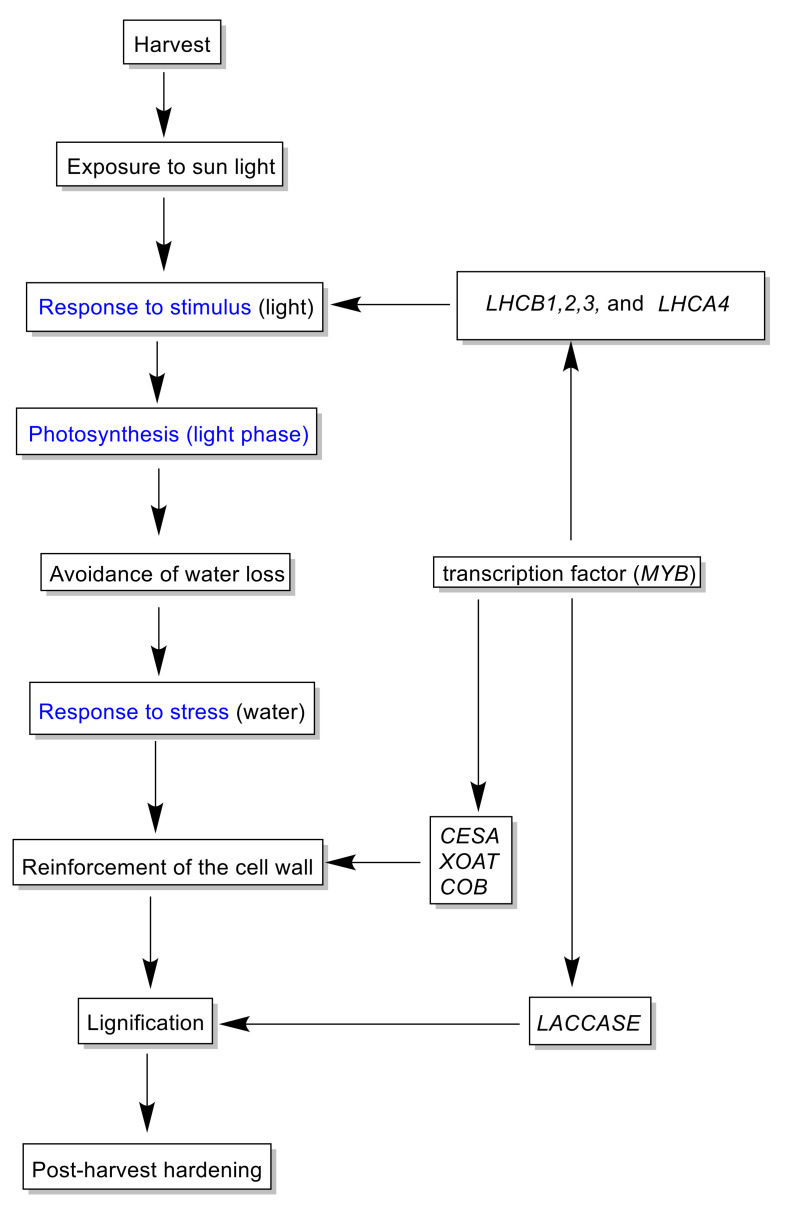
Putative mechanism of the PHH in *D. dumetorum*. Blue represents GO annotation.

**Table 1 plants-10-00787-t001:** Candidate genes associated with post-harvest hardening (PHH) in *D. dumetorum* tuber in the combined analysis of the hardening accessions 3DAH vs. AH.

Contig	LF2C	Padj	Bin/KO	Gene\Name	Description
contig544.g2040	6.91	0.04740	21.4.1.1.3	*FLA*	fasciclin-type AGP
contig278.g50	8.89	0.02720	21.1.2.2	*COB*	regulatory protein
contig760.g29	18.35	0.03609	1.2.3/K05298	*GAPA*	glyceraldehyde-3-phosphate dehydrogenase
contig119.g125	8.07	0.00170	1.1.6.1.1	*PGR5/PGRL1*	complex.component PGR5-like
contig678.g379	7.72	0.00000	1.1.4.1.4/K08910	*LHCA4*	chlorophyll a/b binding protein 4
contig679.g24	7.98	0.00000	1.1.4.1.4/K08910	*LHCA4*	chlorophyll a/b binding protein 4
contig549.g218	6.53	0.00000	K02694	*psaF*	photosystem I subunit III
contig222.g1555	5.27	0.00626	1.1.4.2.8/K02695	*psaH*	photosystem I subunit VI
contig206.g10	5.55	0.00042	1.1.1.1.1/K08912	*LHCB1*	chlorophyll a/b binding protein 1
contig206.g11	7.98	0.00000	1.1.1.1.1/K08912	*LHCB1*	chlorophyll a/b binding protein 1
contig206.g6	7.12	0.00000	1.1.1.1.1/K08912	*LHCB1*	chlorophyll a/b binding protein 1
contig206.g8	7.58	0.00000	1.1.1.1.1/K08912	*LHCB1*	chlorophyll a/b binding protein 1
contig267.g402	5.81	0.00836	1.1.1.1.1/K08913	*LHCB2*	chlorophyll a/b binding protein 2
contig355.g38	5.82	0.01516	1.1.1.1.1/K08913	*LHCB2*	chlorophyll a/b binding protein 2
contig391.g20	6.24	0.00012	1.1.1.1.1/K08912	*LHCB1*	chlorophyll a/b binding protein 1
contig391.g26	6.94	0.00000	1.1.1.1.1/K08912	*LHCB1*	chlorophyll a/b binding protein 1
contig391.g28	5.72	0.00038	1.1.1.1.1/K08912	*LHCB1*	chlorophyll a/b binding protein 1
contig391.g29	7.65	0.00000	1.1.1.1.1/K08912	*LHCB1*	chlorophyll a/b binding protein 1
contig553.g402	4.31	0.04740	1.1.1.1.1/K08914	*LHCB3*	chlorophyll a/b binding protein 3
contig565.g52	7.56	0.02366	1.1.1.1.1/K08912	*LHCB1*	chlorophyll a/b binding protein 1
contig89.g1873	5.94	0.01452	1.1.1.6.2.1	*ELIP*	LHC-related protein group.protein
contig544.g1881	5.17	0.04740	1.1.1.2.13	*1.1.1.2.13/>PsbX*	PS-II complex.component
contig544.g1970	5.05	0.00905	1.1.1.2.13	*1.1.1.2.13/PsbX*	PS-II complex.component
contig267.g494	20.89	0.00000	15.5.2.1/K09422	*MYB*	transcription factor

**Table 2 plants-10-00787-t002:** Candidate genes associated with PHH in *D. dumetorum* tuber in Fonkouankem 1 3DAH vs. AH.

Contig	LF2C	Padj	Bin/Ko	Gene\Name	Description
contig557.g748	9.02	3.15 × 10^−9^	21.1.1.1.1/K10999	*CESA*	cellulose synthase A
contig60.g53	8.86	3.44 × 10^−9^	21.1.1.1.1/K10999	*CESA*	cellulose synthase A
contig73.g5	8.94	3.78 × 10^−9^	21.1.1.1.1/K10999	*CESA*	cellulose synthase A
contig267.g188	23.39	5.99 × 10^−6^	21.1.2.2	*COB*	regulatory protein
contig278.g50	14.51	2.99 × 10^−3^	21.1.2.2	*COB*	regulatory protein
contig143.g88	17.83	2.27 × 10^−3^	21.2.2.2.2	*XOAT*	xylan O-acetyltransferase
contig145.g17	17.90	2.01 × 10^−3^	21.2.2.2.2	*XOAT*	xylan O-acetyltransferase
contig199.g1435	12.17	1.46 × 10^−4^	21.2.2.2.2	*XOAT*	xylan O-acetyltransferase
contig920.g250	17.89	1.76 × 10^−4^	21.2.2.2.2	*XOAT*	xylan O-acetyltransferase
contig922.g12	11.49	7.50 × 10^−3^	21.2.2.2.2	*XOAT*	xylan O-acetyltransferase
contig750.g97	6.45	8.83 × 10^−4^	21.6.1.7/K13066	*COMT*	caffeic acid 3-O-methyltransferase
contig646.g19	5.52	1.68 × 10^−2^	K18368	*CSE*	caffeoylshikimate esterase
contig552.g180	5.18	1.60 × 10^−2^	K00588	*E2.1.1.104*	caffeoyl-CoA O-methyltransferase
contig3.g487	5.66	4.55 × 10^−2^	21.6.1.2/K09754	*CYP98A*	5-O-(4-coumaroyl)-d-quinate 3′-monooxygenase
contig199.g1672	10.14	3.21 × 10^−3^	21.6.2.2/K05909	*E1.10.3.2*	Laccase
contig559.g139	26.23	4.72 × 10^−8^	21.6.2.1	*PMT*	p-coumaroyl-CoA
contig119.g106	14.35	9.30 × 10^−3^	K05350	*bglB*	beta-glucosidase
contig390.g181	6.08	3.53 × 10^−2^	21.3.2.2.2	*BGAL*	beta-galactosidase
contig678.g379	7.74	2.36 × 10^−4^	1.1.4.1.4/K08910	*LHCA4*	chlorophyll a/b binding protein 4
contig679.g24	11.17	4.28 × 10^−6^	1.1.4.1.4/K08910	*LHCA4*	chlorophyll a/b binding protein 4
contig206.g11	7.52	4.19 × 10^−7^	1.1.1.1.1/K08912	*LHCB1*	chlorophyll a/b binding protein 1
contig391.g29	6.83	5.77 × 10^−4^	1.1.1.1.1/K08912	*LHCB1*	chlorophyll a/b binding protein 1
contig546.g79	20.36	6.88 × 10^−4^	15.5.7.2	*DREB*	transcription factor
contig771.g2	25.08	4.05 × 10^−5^	15.5.7.2	*DREB*	transcription factor
contig267.g494	25.57	3.54 × 10^−2^	15.5.2.1/K09422	*MYB*	transcription factor
contig678.g290	16.94	1.44 × 10^−2^	K09422	*MYB*	transcription factor
contig693.g10	6.77	4.76 × 10^−2^	K09422	*MYB*	transcription factor
contig750.g121	25.14	2.61 × 10^−7^	K09422	*MYB*	transcription factor
contig158.g23	37.78	5.01 × 10^−6^	15.5.17	*NAC*	transcription factor
contig556.g459	37.78	5.01 × 10^−6^	15.5.17	*NAC*	transcription factor

## Data Availability

All data in this manuscript is original.

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
