# Peer review of "Transcriptome Sequence Reveals Candidate Genes Involving in the Post-Harvest Hardening of Trifoliate Yam Dioscorea dumetorum"

_plants, 2021, doi:10.3390/plants10040787_

Round 1
Reviewer 1 Report
This manuscript is not written well and has many grammatical and typographical errors. Results and methods are not presented well. Authors identified post-harvest hardening genes need to be functionally validated. I suggest authors to seek help from an English language expert. In addition, I have following concerns/questions:
- In “Results” section line 81-83 seem to be copied format from result section of the journal.
- In fig.1, where conditions are defined, 4DAH should be 14DAH.
- In fig.1 legend – “PCA plot of normalized count using VSD”. What is VSD?
- Different naming - Ibosweet 3 or Ibo sweet 3?
- Line 123-126: Pairwise comparison among 3 stages (AH, 3DAH, 14DAh) is provided but not mentioned that data is from which cultivar?
- Provide the list of all DEGs in all pair-wise comparisons as a supplementary file.
- Line 154-156: “Individual analysis……biological process”. Rephrase the sentence, this sentence makes no sense.
- Line 170: Wrong citation, should be Fig. 5A, B.
- Fig. 4 legend: “Functional annotation……biological process”. Fig.4 A, B show 25 GO terms not top 20. Use “GO terms” instead of “GO pathways”. Y-axis title for Fig. 4 A, B should be “GO terms” rather “Go Name”.
- “Cluster Expression Analysis” - This section makes no sense to me and is very confusing. I suggest elaborating this section and provide appropriate method in Method section with citations.
- “Plant materials”, line 357-360: provide the name of four accessions and their ancestry details. Describe controlled conditions.
- “RNA-seq Extraction”, this heading doesn’t look appropriate. Line 369: provide how many biological and technical replicates included in 48 samples.
- What data normalization read count method was used for differential gene expression analysis?
- Is there any reference for co-expression analysis method, if yes, please provide citation?
Author Response
Response to Reviewer 1 Comments
Point 1: In “Results” section line 81-83 seem to be copied format from result section of the journal.
Response 1: done
Point 2: In fig.1, where conditions are defined, 4DAH should be 14DAH.
Response 2: done
Point 3: In fig.1 legend – “PCA plot of normalized count using VSD”. What is VSD?
Response 3: It was VST: variant stabilizing transformation (VST)
Point 4: Different naming - Ibosweet 3 or Ibo sweet 3?
Response 4: Ibo sweet 3
Point 5: Line 123-126: Pairwise comparison among 3 stages (AH, 3DAH, 14DAh) is provided but not mentioned that data is from which cultivar?
Response 5: Done
Point 6: Provide the list of all DEGs in all pair-wise comparisons as a supplementary file.
Response 6: Done
Point 7: Line 154-156: “Individual analysis……biological process”. Rephrase the sentence, this sentence makes no sense.
Response 7: Done
Point 8: Line 170: Wrong citation, should be Fig. 5A, B.
Response 8: Done
Point 9: Fig. 4 legend: “Functional annotation……biological process”. Fig.4 A, B show 25 GO terms not top 20. Use “GO terms” instead of “GO pathways”. Y-axis title for Fig. 4 A, B should be “GO terms” rather “Go Name”.
Response 9: Done
Point 10: “Cluster Expression Analysis” - This section makes no sense to me and is very confusing. I suggest elaborating this section and provide appropriate method in Method section with citations.
Response 10: Done
Point 11: “Plant materials”, line 357-360: provide the name of four accessions and their ancestry details. Describe controlled conditions.
Response 11: Done
Point 12: “RNA-seq Extraction”, this heading doesn’t look appropriate. Line 369: provide how many biological and technical replicates included in 48 samples.
Response 12: Done
Point 13: What data normalization read count method was used for differential gene expression analysis?
Response 13: DESeq2 offers two transformations for count data that stabilize the variance across the mean. We used the variance stabilizing transformation (VST).
Point 14: Is there any reference for co-expression analysis method, if yes, please provide citation?
Response 14: Here is the reference used for co-expression analysis method: https://2-bitbio.com/2017/10/clustering-rnaseq-data-using-k-means.html
Reviewer 2 Report
The study presented in the manuscript refers to the transcriptomic analysis of 4 genotypes in yams (Dioscorea dumetorum) in four time point. The aim of the study was to identified genes involved in post-harvesting hardening (PHH). The authors conducted a comparison of RNA-seq results and identified DEGs. Classically like in this type of research, they made also GO and KEGG analysis. As the result, the authors proposed some genes that could be related to the hardening of tubers. They also constructed a putative mechanism of the PHH. This is the first report of this type of research on this plant species. I believe that the results could shed light on the elucidation of molecular mechanisms of PHH and benefit future improvements in the breeding of yams.
The work has a typical layout for this type of research. The authors undoubtedly put a lot of effort into research. The results are quite well described and discussed, and the conclusions drawn are logical. The whole manuscript is good written, clear and easy to follow. However, I have a few comments about its preparation.
Here is a list of my comments:
- At the beginning of the results section, there is most likely a text from the form informing you what the results should be - it needs to be removed.
- The authors took 4 yams genotypes for the study but did not describe what they are characterized by or differ. This type of description of the research material should also be placed at the beginning of the results to make it easier for readers to understand the meaning of the research. A brief description of the experiment would be appropriate here.
- in section 2.1 the results, there is no reference to which reference genome (from which genotype) the transcriptomes were mapped - please complete this.
- line 321 - the sentence should not begin with the number of the citation, it is better to start the sentence with the name of the author of the quoted article.
- there is a typo in figure 1: there is 4DAH, but it should be 14DAH,
- in the materials and methods section 4.1, the description of the source of the plant material is too general, it should be described in more detail.
- line 364 a typo - there's a "9" next to ME.
- subsection 4.2: description of storage until the 3rd and 14th day "AH" is incomplete - it should be supplemented (temperature, light, etc.). Especially since the authors suggest the influence of light on the PHH process.
Author Response
Response to Reviewer 2 Comments
Point 1: At the beginning of the results section, there is most likely a text from the form informing you what the results should be - it needs to be removed.
Response 1: done
Point 2: The authors took 4 yams genotypes for the study but did not describe what they are characterized by or differ. This type of description of the research material should also be placed at the beginning of the results to make it easier for readers to understand the meaning of the research. A brief description of the experiment would be appropriate here.
Response 2: done
Point 3: in section 2.1 the results, there is no reference to which reference genome (from which genotype) the transcriptomes were mapped - please complete this.
Response 3: done
Point 4: line 321 - the sentence should not begin with the number of the citation, it is better to start the sentence with the name of the author of the quoted article.
Response 4: done
Point 5: there is a typo in figure 1: there is 4DAH, but it should be 14DAH,
Response 2: done
Point 6: in the materials and methods section 4.1, the description of the source of the plant material is too general, it should be described in more detail.
Response 2: done
Point 7: line 364 a typo - there's a "9" next to ME.
Response 2: done
Point 8: subsection 4.2: description of storage until the 3rd and 14th day "AH" is incomplete - it should be supplemented (temperature, light, etc.). Especially since the authors suggest the influence of light on the PHH process.
Response 2: done
Round 2
Reviewer 1 Report
This improved version of manuscript looks good, however minor spelling and grammar check is required. Good luck!
Author Response
Checked.